# The Basic Mechanical Properties and Shrinkage Properties of Recycled Micropowder UHPC

**DOI:** 10.3390/ma16041570

**Published:** 2023-02-13

**Authors:** Chengfang Yuan, Yang Chen, Dongxu Liu, Weiqian Lv, Zhe Zhang

**Affiliations:** 1Yellow River Laboratory, Zhengzhou University, Zhengzhou 450001, China; 2College of Water Resources and Civil Engineering, Zhengzhou University, Zhengzhou 450001, China; 3Henan Provincial Communications Planning & Design Institute Co., Ltd., Zhengzhou 450018, China; 4Henan Province A.L Expressway Construction Co., Ltd., Zhengzhou 450016, China

**Keywords:** recycled micropowder, UHPC, mechanical property, autogenous shrinkage, drying shrinkage, shrinkage rate prediction mode

## Abstract

Using waste clay brick powder (RBP) to partially replace cement in the preparation of concrete, is one way to recycle construction waste. In order to investigate the physical and mechanical properties and volume stability of recycled micropowder ultra-high-performance concrete (UHPC), the basic mechanical and shrinkage properties of recycled micropowder UHPC were studied at replacement rates of 10%, 20%, 30%, 40% and 50%. The results show that: (1) When the activated recycled brick powder is used to replace the cement, the compressive strength, flexural strength and splitting tensile strength of the UHPC initially increase and then decrease with the increase in the substitution rate. When the substitution rate is 10%, the 28 d compressive strength, flexural strength and splitting tensile strength of the UHPC are the highest; (2) Replacing cement with recycled brick powder can reduce the autogenous shrinkage of the UHPC, and the autogenous shrinkage rate of the UHPC decreases with the increase in the brick powder replacement rate. The drying shrinkage of the UHPC can be reduced by replacing cement with recycled brick powder. The drying shrinkage of the UHPC initially decreases, and then increases, with the increase in the replacement rate of brick powder. When the replacement rate of the brick powder was 30%, the drying shrinkage of the UHPC was the least, and this was 49.7% lower than that in the benchmark group. The prediction models of autogenous shrinkage and drying shrinkage are in good agreement with the experimental results, which can be used to predict the shrinkage development of recycled brick powder UHPC.

## 1. Introduction

The continuous social and economic development of the construction industry has meant that a large amount of construction waste is generated globally, and this is expected to remain high over the next decades [1,2,3,4]. The United States and the European Union generate about 700 million tons and 800 million tons of C&D waste, respectively, each year, and China’s C&D waste exceeds 1.8 billion tons each year [5,6]. By the end of 2020, China’s construction waste stockpile had reached about 20 billion tons, yet the utilization rate was less than 10% [7,8]. The traditional extensive stacking and landfill of construction waste not only occupies a large amount of land, but also has environmental hazards [9,10,11,12]. The recycling and reuse of construction waste has been proposed as an effective and sustainable alternative [13]. Studies have shown that about 80% of construction waste is reusable, such as waste concrete and bricks [2,12]. At present, recycled concrete aggregates have been widely used in practical structural engineering [14,15]. Since the porosity of brick aggregates is much higher than that of concrete aggregates, limiting the potential use of brick aggregates in concrete products, a viable method is needed to recycle brick waste [16]. Relevant research shows that clay brick micropowder has certain activity, and it can be used as an auxiliary cementing material for concrete production after it is stimulated in a physical or chemical way, which provides a new idea for the resource utilization of waste bricks [17,18,19]. Ultra-high-performance concrete (UHPC) is a cement-based composite material with ultra-high strength and excellent durability [20,21,22]. However, the large amount of cement and high cost of UHPC also affect the application of these materials to some extent. The use of brick powder to partially replace cement in the preparation of recycled ultra-high-performance concrete not only reduces the amount of cement, costs and energy consumption, but also opens up a new path for the green development of UHPC.

It is well known that physical and mechanical properties and volume stability are important aspects of concrete performance [23,24,25,26]. The mechanical properties and shrinkage properties of recycled brick powder cement-based materials have been reported. Zhiming Ma et al. [27] used waste brick powder (WBP) to prepare environmentally friendly mortar and found that the addition of WBP reduced the drying shrinkage. When the fineness of WBP is higher than that of cement, the compressive strength increases with the increase in WBP content, reaching 15%. When the fineness of WBP is close to or lower than that of cement, the compressive strength decreases with the increase in the WBP substitution ratio. Zhi-hai He et al. [28] used recycled brick powder (RBP) to replace part of the silicon powder (SF) and discussed the effect of the RBP content on the strength and autogenous shrinkage of the UHPC material. The results show that the strength of the material can be reduced by using 30–45% RBP instead of SF, but the strength of the material can be significantly improved by using 15% RBP. Meanwhile, the autogenous shrinkage of the UHPC decreased significantly with the increase in the RBP replacement rate. Huixia Wu et al. [29] studied the micro- and macro-properties of green mortar containing various WCBP and found that the use of WCBP reduced the compressive strength of green mortar. Mortar containing waste brick powder has better compressive strength than mortar containing waste concrete powder. Mahmood Anwar S A et al. [30] used clay brick powder (CBP) to partially replace cement to prepare concrete and conducted a 28-day strength test. It was found that due to the high Al_2_O_3_ and SiO_2_ contents, C-A-S-H and C-S-H gels were formed with 5% brick powder, which improved the bending and compressive strength of the materials, while the use of a higher percentage of brick powder resulted in a decrease in the strength of the materials. Rovnanik et al. [31] examined the use of brick waste in geopolymers and reported that the compressive strength of a low-calcium fly ash-based geopolymer was reduced with over 25% brick powder as a binder replacement. Ahmed M F et al. [32] prepared metakaolin-based geopolymer concrete (MK-Gpc) using clay brick waste as a cementing material and evaluated its influence on the shrinkage properties of materials. The results show that 10–30% clay brick powder can improve the drying shrinkage of the material.

Existing studies show that the mechanical and shrinkage properties of cement-based materials can be improved by replacing cement partially with an appropriate amount of clay brick powder. Based on the above research reports, this paper reports a study of the compressive, flexural, and splitting tensile properties, and the autogenous and drying shrinkage properties of recycled brick powder UHPC, with the replacement rate of recycled brick powder as the main variable. Based on the experimental results, a shrinkage prediction model of recycled brick powder UHPC is proposed, which provides an experimental basis and theoretical guidance for the application of recycled brick powder UHPC.

## 2. Materials and Methods

### 2.1. Raw Material

This test used the Mengdian Group Cement Co., Ltd., Xinxiang City, Henan Province, China. P.O 52.5 ordinary Portland cement; its main technical indicators are shown in Table 1, and the apparent density of cement is 3100 kg/m^3^. The admixture was first-class fly ash from the Yuanheng Environmental Protection Engineering Co., Ltd., Zhengzhou City, Henan Province, China.—the apparent density of first-class fly ash is 2363 kg/m^3^, and silica ash with a SiO_2_ content equal to 96.76%—the apparent density of silica ash is 2200 kg/m^3^. The recycled brick powder came from the sintered waste clay bricks resulting from demolition in Zhengzhou City. The compressive strength of the waste bricks was MU20~MU25, and the brick powder used in the test was obtained by jaw crushing, screening, and ball milling (ball milling 45min). The preparation process is shown in Figure 1. The main material property indicators and components of recycled brick powder are shown in Table 2. Natural river sand with a particle size of 0.075~1.25 mm was screened out as an aggregate. The fiber was made of Zhitai brand copper-plated fine steel fiber, with length of 12 mm, diameter of 0.2 mm, density of 7.8 g·cm^−3^, and tensile strength of more than 2000 MPa. Using a CQJ-JSS type polycarboxylic acid high-efficiency water-reducing agent produced by Shanghai Chenqi Chemical Technology Co., Ltd., China. the water reduction rate was 32%. The water for test mixing and curing was the ordinary tap water of Zhengzhou City.

### 2.2. RBP Preparation and Property Determination

The recycled brick powder comes from the sintered waste clay bricks produced by demolition in Zhengzhou city; the compressive strength of the waste bricks was MU20~MU25. Figure 1 shows the preparation process of recycled brick powder. Bricks were initially selected from construction waste, then crushed by a jaw crusher and sifted into a 75 μm powder. After 45 min of ball milling, the recycled brick powder was activated and finally 32.8 μm was obtained, which is similar to cement fineness. The particle size distribution is shown in Figure 2 and Figure 3. Generally, a very fine recycled brick powder contributes to the activity and filler effect in its pozzolanic cement composites. The main indicators and components of recycled brick powder are shown in Table 2.

The influence of the recycled brick powder on the properties of cement-based materials is determined by many factors, which are not only related to the specific surface area of the materials, but also closely related to the morphology of recycled brick powder, the interaction of recycled brick powder with other materials and other factors. The cement and brick powder were placed in a vacuum drying oven, vacuum dried at 50 °C for 1 h, and subsequently, sealed and stored for scanning electron microscopy. Figure 4a,b are the images of cement and brick powder magnified 1000 times. From Figure 4a, it can be seen that the size and shape of cement particles are relatively consistent. Occasionally, several particles stick together to form a relatively large cluster. Compared with Figure 4b, it is evident that the overall particle size of brick powder after physical excitation (ball milling for 45 min) is smaller than that of cement particles, the shape is more irregular and the specific surface area is larger. These fine brick powder particles are easier to wrap over the surface of the cement, which shows, from a side angle, that too much brick powder will hinder cement hydration. In Figure 4b, a single brick powder particle larger than 50 μm can be seen on the left, and there are many fine holes on the surface. Due to the existence of some such particles, a small amount of water used for the cement hydration reaction will be absorbed in the early stage, which may cause the initial strength to decrease. The degree of reaction between the brick powder and the water is very low. Subsequently, the difference in relative humidity around the brick powder inside the UHPC releases this part of the absorbed water again, forming an internal curing effect, which may lessen the 28 d strength more in the benchmark group than the substitution rate group. This makes it possible to partially replace cement with recycled brick powder as cementing material.

The reclaimed micropowder was milled by a SM500×500 ball mill, after which the mortar specimens were prepared by replacing cement with a 30% content, and an XRD analysis was carried out. Figure 4 shows that the sample mainly contained SiO_2_, Ca(OH)_2_, CaCO_3_ and C-S-H. In Figure 5, the diffraction peak of the Ca_3_Al_2_O_6_ crystal may result because the recycled micropowder was not ball milled and contains some cement mortar. The Ca_3_Al_2_O_6_ crystal may be the product of secondary hydration. The SiO_2_ in the sample after ball milling is mainly from sand and recycled fine powder in the aggregate, and the Ca(OH)_2_ is mainly from cement hydration. This indicates that after mechanical ball-milling excitation, the SiO_2_ in some recycled micropowders is activated and reacts with the hydration product of cement, Ca(OH)_2_, to form a C-S-H gel, which helps to improve the strength of the materials. The recycled brick powder is prepared from waste clay bricks, with silicon accounting for most of the composition of the clay bricks. A previous investigation reported that the high content of silicon and its oxide contained in RBP contribute to the pozzolanic reaction of cementitious materials [33]. However, because no hydrated component is contained in RBP, incorporating a high-volume of RBP into cement composites is not recommended.

### 2.3. Test Mix Proportion

The standard UHPC with a designed water–binder ratio of 0.17 is used to replace cement with 10%, 20%, 30%, 40% and 50% mass substitution rates to prepare recycled brick powder UHPC. The details of mixes are listed in Table 3.

### 2.4. Experimental Design

#### 2.4.1. The Mechanical Properties Test Design

The specimens were designed and tested according to T/CCPA 7-2018 (T/CBMF 37) [34] and (GB/T 50081-2019) [35]. The cubic compressive strength test was conducted using a YAW-2000B pressure testing machine. The loading speed was 0.8 MPa/s, and the loading was continuous and uniform until the specimen was crushed. The load value at this point was recorded, and the final compressive strength value was calculated from the load value. The bending strength test used the WDW-100 electronic universal testing machine. The loading speed was 0.08 MPa/s, and the loading was continuous and uniform until the specimen was crushed. The load value at crushing was recorded and the bending strength value was calculated. The splitting tensile strength test used a YAW-2000B pressure testing machine—the loading rate and process was the same as the compressive strength test. The failure load was recorded at the time of crushing the specimen, and the splitting tensile strength was calculated from recorded load.

#### 2.4.2. The Shrinkage Test Design

The specimens were designed and tested according to the T/CCPA7-2018(T/CBMF37) [34], (DB13/T 2946-2019) [36] and (GB/T 50082-2009) [37]. The autogenous shrinkage test used the NEL-NES non-contact concrete shrinkage deformation tester. The contact method is used for drying shrinkage, and the sp-540 horizontal concrete shrinkage dilatometer is used. The size of the test block was 100 mm × 100 mm × 515 mm. The test is shown in Figure 6 and Figure 7. During the test, the steel bracket placed in contact with the target was placed in the test mold in advance and buried when the test block was formed. Immediately following the pouring, a layer of film was placed on the surface of the test block to maintain freshness, after which the block was moved into a constant temperature and humidity chamber with a temperature of 20 ± 2 ℃ and a relative humidity of 60 ± 5%. The autogenous shrinking test block shall be kept for 3 hours at constant temperature and humidity, and the data shall be collected every 15 minutes. After the dry shrinkage test block is poured, it is moved to the standard curing room for curing with formwork for 2 days, and then the formwork is removed, and it is moved to the constant temperature and humidity room for testing at the age of 3 days. And its shrinkage was tested at the following intervals: 1 d, 3 d, 7 d, 14 d, 28 d, 42 d and 56 d (calculated from the time when the test piece was moved into the constant temperature and humidity chamber). It was essential to avoid the vibration of the test device during the test.

## 3. Results and Discussion

### 3.1. Compressive Strength Test

The UHPC compressive strength test results of recycled brick micropowder under different replacement rates are shown in Figure 8 and Table 4.

From Figure 8 and Table 4, it is evident that at the age of 28 d, the compressive strength of the UHPC, after replacing cement with recycled brick micropowder, was lower than that in the benchmark group. When the replacement rate was 10%, 20%, 30%, 40% and 50%, the 28 d compressive strength of the UHPC was 141.3MPa, 135.2MPa, 133.6MPa, 124.7 MPa and 123.1 MPa, respectively, which was 5.7%, 9.8%, 10.9%, 16.8% and 17.9% lower, respectively, than that in the benchmark group. As the age increased, the compressive strength of the recycled brick micropowder UHPC showed an overall increase. The greater the replacement rate, the less obvious the trend. The compressive strength of the UHPC with a 50% replacement rate was almost the same at each age. At 7d, 14d and 28 d, the compressive strength of the UHPC at the 0% replacement rate in the benchmark group was 131.0 MPa, 139.1 MPa and 149.9 MPa, respectively, which is 8.4%, 15.1% and 24.1% higher, respectively, than the compressive strength at 3d. At a replacement rate of 50%, the 28 d compressive strength of the recycled brick micropowder UHPC was 123.1 MPa, which was only 2.9% higher than that at 3d. At a replacement rate of 10%, the compressive performance of the recycled micropowder UHPC was the best. It is worth noting that the 28 d compressive strength of each group of the recycled brick micropowder UHPC specimens was greater than 120 MPa, which meets the provisions for UHPC compressive strength in the T / CCPA 7-2018 (T/CBMF 37) [34]. As the replacement rate of the recycled brick powder increased, the compressive strength of the material showed a downward trend as a whole. This was because a large proportion of the RBP significantly reduced the hydration products, resulting in a reduction in the compressive strength of the material [28]. The older the age, the more significant the downward trend.

### 3.2. The Flexural Strength Test

The flexural strength test results of the recycled brick micropowder UHPC with different replacement rates are shown in Figure 9 and Table 5.

From Figure 9 and Table 5, it is evident that at the age of 28 d, the flexural strength of the UHPC, after having replaced the cement with recycled brick micropowder, was lower than that in the benchmark group. When the replacement rate was 10%, 20%, 30%, 40% and 50%, the 28 d flexural strength of the UHPC was 25.0 MPa, 22.4 MPa, 20.7 MPa, 19.1 MPa and 18.1 MPa, respectively, which was 2.7%, 12.8%, 19.5%, 25.7% and 29.6% lower, respectively, than that in the benchmark group. With the increase in age, the flexural strength of the recycled brick micropowder UHPC showed an increasing trend as a whole. The larger the replacement rate, the less obvious the trend. The flexural strength of the UHPC with a 50% replacement rate was almost the same at each age. At 7d, 14d and 28 d, the flexural strength of the UHPC in the benchmark group, which had a 0% replacement rate, was 21.8 MPa, 23.5 MPa and 25.7 MPa, respectively, which was 9.0%, 17.5% and 28.5% higher, respectively, than that at 3d. When the substitution rate was 50%, the 28 d flexural strength of the recycled brick micropowder UHPC was 18.1 MPa, which was only 6.5% higher than that at 3d. When the replacement rate was 10%, the flexural performance of the recycled micropowder UHPC was the best. It is worth noting that the 28 d flexural strength of each group of the recycled brick micropowder UHPC specimens was greater than 15 MPa, which meets the provisions for UHPC flexural strength in the T/CCPA 7-2018 (T/CBMF 37) [34]. As the replacement rate of the recycled brick powder increased, the flexural strength of the material showed a downward trend. The older the age, the more significant the downward trend. [38,39].

### 3.3. The Splitting Tensile Strength Test

The splitting tensile strength test results of the recycled brick micropowder UHPC with different replacement rates are shown in Figure 10 and Table 6.

As is evident from Figure 10 and Table 6, at the age of 28 d, the UHPC splitting tensile strength after the recycled brick micropowder replaced the cement was lower than that in the benchmark group. With the increase in the replacement rate of recycled brick micropowder, the splitting tensile strength of the recycled brick micropowder UHPC showed a downward trend as a whole. The older the age, the more significant the downward trend, while the splitting tensile strength of the UHPC at 3d of age was almost the same. When the substitution rates were 10%, 20%, 30%, 40% and 50%, the 28 d splitting tensile strength of the UHPC was 13.77 MPa, 13.43 MPa, 13.18 MPa, 12.78 MPa and 11.50 MPa, respectively, which was 1.2%, 3.7%, 5.5%, 8.3% and 17.5% lower, respectively, than in the benchmark group. With the increase in age, the splitting tensile strength of the recycled brick micropowder UHPC increased as a whole. The greater the replacement rate, the less obvious the trend. The splitting tensile strength of the UHPC with a 50% replacement rate increased the least. When the age was 7d, 14d and 28 d, the splitting tensile strength of the UHPC at the 0% replacement rate in the benchmark group was 12.24 MPa, 13.09 MPa and 13.94 MPa, respectively, which was 12.3%, 20.1% and 27.9% higher, respectively, than that at 3d. When the replacement rate was 50%, the 28 d splitting tensile strength of the recycled brick micropowder UHPC was 11.50 MPa, which was only 15.2% higher than that at 3d. When the replacement rate was 10%, the splitting tensile performance of recycled micropowder UHPC was the best.

The above experimental results show that the replacement rate of recycled brick micropowder had little effect on the initial strength of the UHPC. When the replacement rate increased, the subsequent strength of the UHPC with recycled brick micropowder was less. The compressive strength, flexural strength and splitting tensile strength of the UHPC with recycled brick micropowder were lower than those of the benchmark UHPC with a zero replacement rate. However, after active excitation the material properties of the recycled brick powder were still inferior to the performance of cement. The extremely low water–binder ratio of the UHPC resulted in a large amount of cement in the system with a low degree of hydration, which only played a physical filling role, the proportion of unhydrated particles was large, and the material utilization rate was low. In addition, the water absorption characteristics of recycled brick powder reduced the actual water–binder ratio of the system and compensated for part of the loss of compressive strength and splitting tensile strength [40]. The flexural strength is mainly related to the steel fiber. Moreover, when the strength of the matrix decreases, the ductility of the concrete is relatively improved, which compensates for part of the flexural strength loss. Therefore, when the recycled brick powder replacement rate is low (such as 10%), the strength is basically the same as the zero-replacement rate in the benchmark group. With the increase in the replacement rate of recycled brick micropowder, the addition of too much recycled brick micropowder will absorb a large amount of free water for mixing, which leads to the failure of the cementitious materials, such as cement clinker particles to cover the fine aggregates, such as sand, and cannot effectively form a cemented system, resulting in reduced compressive and splitting tensile strengths [17]. In such cases, the binding between the matrix and the steel fiber is not tight enough, and the “bridging” effect of the steel fiber gradually fails, thus weakening the ultimate bearing capacity of the UHPC specimen in the flexural test. As the age increased, the compressive strength of the UHPC with different substitution rates gradually increased, indicating that the activated recycled brick micropowder contained some active silica, showing pozzolanic activity [41,42].

### 3.4. Autogenous Shrinkage Test

The Meridional curve of the autogenous shrinkage rate of the recycled brick micropowder UHPC is shown in Figure 11.

Figure 11 shows that the development law of the autogenous shrinkage rate of each group of specimens is roughly divided into four stages. The first stage is 0~6 h. At this stage, the cement hydration reaction is intense, the skeleton has not yet been formed inside the UHPC, and new substances are continuously generated inside the slurry, the water is continuously consumed, the material volume is reduced, the slope of the shrinkage rate curve is the largest, and the shrinkage is relatively rapid. The second stage is 6 h~72 h. At this stage, the slope of the shrinkage curve of the UHPC decreases, and the skeleton begins to form inside the UHPC, which reduces the development of shrinkage to a certain extent, and the shrinkage growth begins to slow down. The third stage is 72 h~84 h, that is, 3d~3.5d after pouring. At this time, the UHPC has been condensed and hardened. With the hydration reaction, water is continuously lost from the capillary pores of the material and forms a meniscus. The surface tension of the water in the meniscus causes the cement matrix to shrink, and the shrinkage increases slightly at this time. The fourth stage is 84 h~200 h, when the UHPC has been completely hardened, resulting in resistance to the autogenous shrinkage strength, the autogenous shrinkage curve tends to be stable.

In the present study, when the replacement rate increased from 0% to 10%, the autogenous shrinkage of the UHPC decreased obviously. At 168 h (7d), the shrinkage rate of the R1 group was 37.5% lower than that in the benchmark group. When the replacement rate increased from 10% to 30%, the autogenous shrinkage development law of the UHPC was relatively close. After the shrinkage was basically stable, the shrinkage rate of R1, R2 and R3 was not much different. The 168 h shrinkage rate of R2 and R3 was reduced by 36.6% and 37.2%, respectively, compared with the benchmark group. When the replacement rate was 40–50%, the shrinkage rate of the UHPC continued to decrease, and the 168 h-autogenous shrinkage rate of R4 and R5 was 61.2% and 63.7% lower, respectively, than that in the benchmark group. The results show that the autogenous shrinkage of each group of specimens decreased with the increase in the replacement rate of brick powder. This was because the addition of the RBP reduced the cement composition, thereby reducing the autogenous shrinkage value of the UHPC [28,43].

### 3.5. The Drying Shrinkage Test

The drying shrinkage time curve of the recycled brick powder UHPC is shown in Figure 12.

Figure 12 shows that the drying shrinkage of each group of test pieces developed rapidly before the age of 14 days, and slowly after the age of 14 days. At the age of 42 days, the drying shrinkage of the UHPC was basically stable. With the increase in the replacement rate of the recycled brick powder, the drying shrinkage rate of each group decreased continuously. Compared with the baseline group, the 56d drying shrinkage rate of the R1 to R5 groups decreased by 21.9%, 31.8%, 49.7%, 42.7% and 45.6%, respectively. The drying shrinkage of the material decreased significantly after the recycled brick powder replaced the cement. The loss of water and the increase in recycled brick powder reduced the drying shrinkage of the materials [44].

According to the above test results, the 28 d drying shrinkage rates of R0~R5 account for 93.1%, 90.9%, 89.0%, 88.4%, 92.6% and 91.5% of the whole test cycle, respectively. It is evident that the drying shrinkage of the reclaimed brick powder UHPC mainly occurs before the age of 28 d. The shrinkage of the UHPC is mainly autogenous shrinkage and drying shrinkage. The drying shrinkage of R1~R5 accounts for 18.5%, 22.1%, 19.6%, 15.4%, 17.2% and 26.4% of the total shrinkage, respectively. When the replacement rate was 30%, the drying shrinkage of the UHPC reached the minimum. The drying shrinkage of the recycled brick powder UHPC accounted for about 15–26% of the total shrinkage.

## 4. The Shrinkage Prediction Model of Recycled Micropowder UHPC

### 4.1. The Autogenous Shrinkage Model of the Recycled Brick Powder UHPC

The research on the shrinkage model at home and abroad mostly focuses on an ordinary UHPC, and the existing model is not fully applicable to the autogenous shrinkage prediction of the recycled brick powder UHPC due to different test conditions, the material mix ratio, specimen size and test methods. Therefore, the autogenous shrinkage model of the recycled brick powder UHPC is established based on the experimental results of this paper, as shown in Formula (1):(1)$$\varepsilon_{c}(t)=-\delta \exp(\frac{-t}{a})+b$$
(2)$$\delta =2050-43.97r+0.93r^{2}-0.01r^{3}$$
(3)$$a=43.83+(\frac{754}{15.07\sqrt{\frac{\pi }{2}}})*\exp[-2{\left(\frac{r-23.07}{15.07}\right)}^{2}]$$
where, ε*_c_(*t*)* is the autogenous shrinkage value, 1 × 10^−6^; δ and a are the influence coefficients of brick powder, which are calculated by Formula (2) and Formula (3), respectively. b is the final strain value of autogenous shrinkage of each specimen; t is the test age, *h*; r is brick powder replacement rate, *%*.

The calculation results of the prediction model are compared with the experimental data to verify the accuracy of the model. The benchmark group and the 40% substitution rate group are selected for analysis, and the results are shown in Figure 13.

Figure 13 shows that there is a deviation between the predicted curve and the test curve between 0~6 h. This is because the specimen has not been initially set before 6h, and the autogenous shrinkage change value is large. With the increase in time, the prediction curve and the test curve gradually overlap, and the coincidence is good. Formula (3) can be used as a prediction model for the autogenous shrinkage of the recycled brick powder UHPC.

### 4.2. Drying Shrinkage Model of Recycled Brick Powder UHPC

The drying shrinkage prediction model of recycled brick powder UHPC was obtained by fitting the drying shrinkage test data, as shown in Equation (4):(4)$$\varepsilon_{ds}(t)=m+\frac{m}{1+{\left(\frac{t}{n}\right)}^{p}}$$
(5)$$n=7.17+0.08r-4.19r^{2}$$
where: ε*_ds_*(t) is drying shrinkage, 1 × 10^−6^; m is a constant, and its value is the final drying shrinkage value of each group of specimens. t is the test age, *d*; n is the influence coefficient of the replacement rate of brick powder, which is calculated according to Equation (5). r is the replacement rate of brick powder, %; *p* is a constant, (when r ≦ 30, *p* is 1.5; when r > 30, *p* is 2.0). Comparing the prediction model calculation results with the drying shrinkage test results, the benchmark group and 40% substitution rate group are still selected for analysis, and the results are shown in Figure 14. The predicted curve is in good agreement with the experimental curve, and Equation (4) can be used as the drying shrinkage prediction model of recycled brick powder UHPC.

## 5. Conclusions

In this paper, the compressive, flexural and splitting tensile properties, and the autogenous and drying shrinkage properties of the recycled brick powder UHPC were studied with the substitution rate of recycled brick powder as a variable. Based on the test results, a shrinkage prediction model of the reclaimed brick powder UHPC was proposed. The results are as follows:(1)The compressive strength of the UHPC decreased with the increase in replacement rate after the cement was replaced by activated recycled brick powder, and the decreasing trend was more obvious with the older age. An increase in age resulted in the compressive strength of the recycled brick micropowder UHPC showing an increasing trend. At the age of 28 days, the compressive strength of the UHPC in each group after replacement was lower than that in the benchmark group. When the replacement rate of recycled brick powder was 10%, the 28 d compressive strength was the highest at 141.3 MPa;(2)The flexural strength of the UHPC decreased with the increase in replacement rate after the cement was replaced by activated recycled brick powder, and the decreasing trend was more obvious with age. As age increased, the flexural strength of the recycled micro-brick powder UHPC showed an increasing trend. At the age of 28 days, the flexural strength of the UHPC in each group after replacement was lower than that in the benchmark group. When the replacement rate of the recycled brick powder was 10%, the 28 d flexural strength was the highest at 25.0 MPa;(3)The splitting tensile strength of the UHPC decreased with the increase in replacement rate after the cement was replaced by activated recycled brick powder, and the decreasing trend was more obvious with age. Alternatively, with the increase in age, the splitting tensile strength of the recycled brick micropowder UHPC showed an increasing trend. At the age of 28 days, the splitting tensile strength of the UHPC in each group, after replacement, was lower than that in the benchmark group. When the replacement rate of recycled brick powder was 10%, the 28 d splitting tensile strength was the highest at 13.77 MPa;(4)The autogenous shrinkage of the recycled brick powder UHPC developed most rapidly in the first 0–6 h. After the recycled brick powder replaced the cement, the autogenous shrinkage of the UHPC decreased, and with the increase in the replacement rate, the autogenous shrinkage of the UHPC decreased;(5)The drying shrinkage of the UHPC can be reduced after the recycled brick powder replaces the cement, and with the increase in the replacement rate of recycled brick powder, the drying shrinkage of the UHPC presented a trend of initially decreasing and then increasing. When the replacement rate of the brick powder was 30%, the drying shrinkage of the UHPC was the smallest, which was 49.7% lower than that in the benchmark group. The drying shrinkage of the recycled brick powder UHPC mainly occurred in the first 28 d, and the drying shrinkage at 28 d accounts for about 90% of the drying shrinkage value of the whole test span.(6)The autogenous shrinkage and age of the recycled brick powder UHPC conform to the exponential function relationship based on *e*, and the drying shrinkage and age conform to the exponential function relationship. The autogenous shrinkage prediction model and drying shrinkage prediction model of the recycled brick powder UHPC are in good agreement with the test results, which can be used to predict the shrinkage of recycled brick powder UHPC.

## Figures and Tables

**Figure 1 materials-16-01570-f001:**
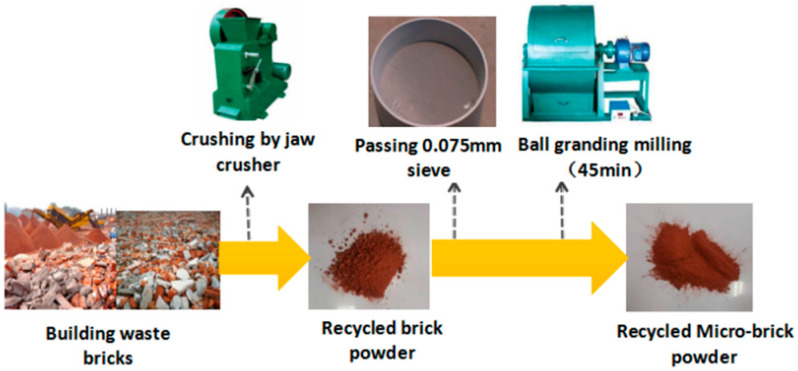
A flow chart of recycled brick powder preparation.

**Figure 2 materials-16-01570-f002:**
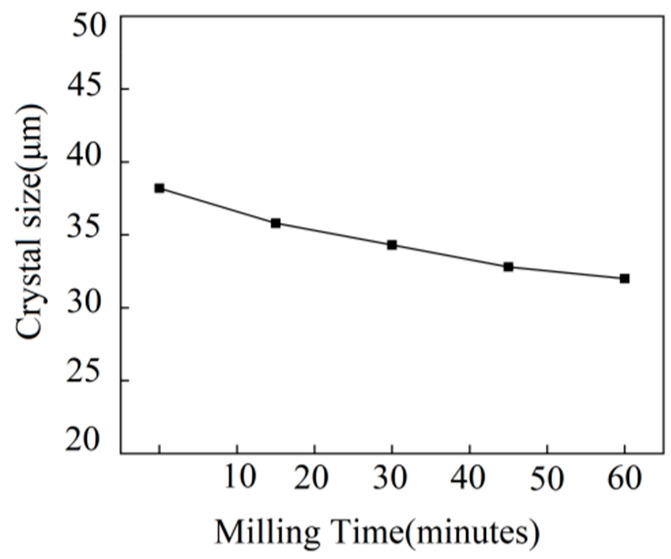
The characterization of brick powder.

**Figure 3 materials-16-01570-f003:**
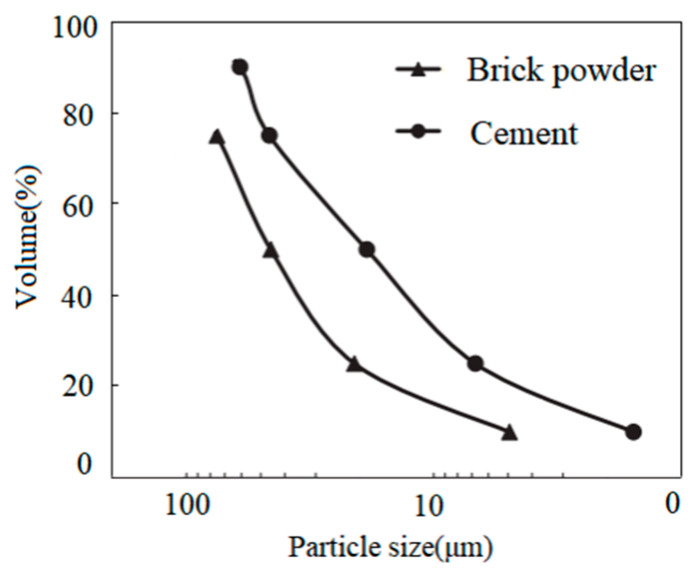
The size distribution of cement and RBP.

**Figure 4 materials-16-01570-f004:**
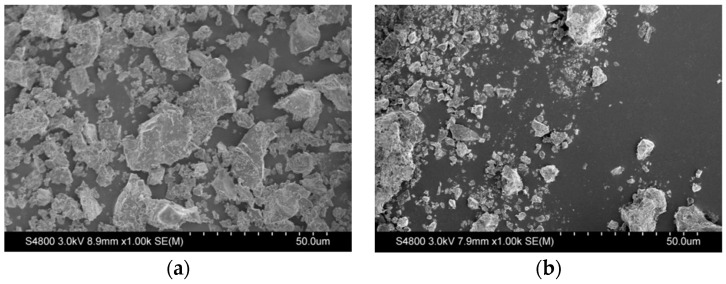
A micrograph of cement and recycled brick powder. (**a**) Cement; (**b**) Recycled brick powder.

**Figure 5 materials-16-01570-f005:**
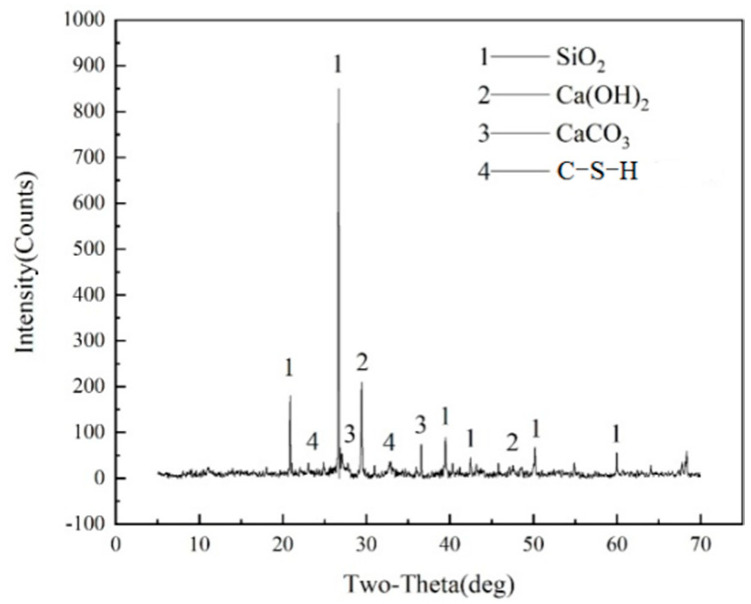
An XRD analysis of activated recycled brick mortar.

**Figure 6 materials-16-01570-f006:**
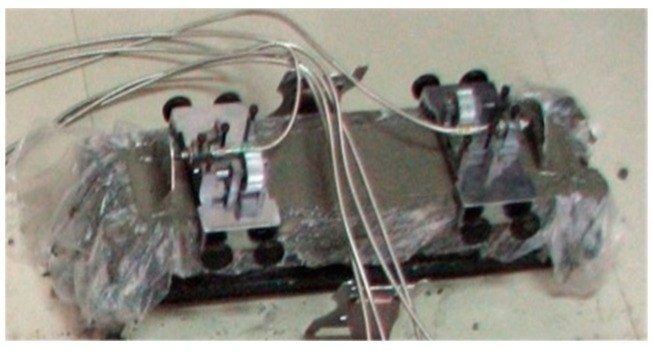
Autogenous shrinkage test.

**Figure 7 materials-16-01570-f007:**
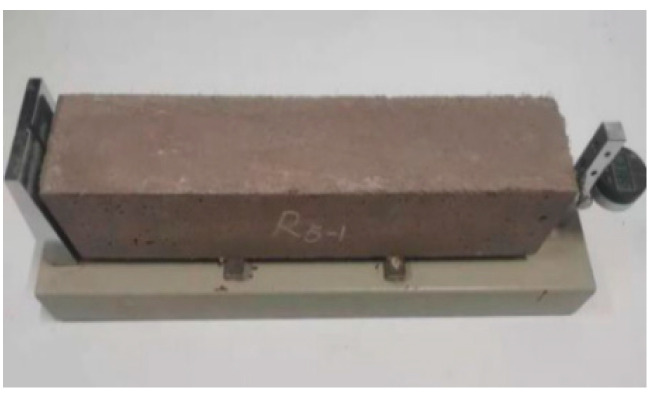
Drying shrinkage test.

**Figure 8 materials-16-01570-f008:**
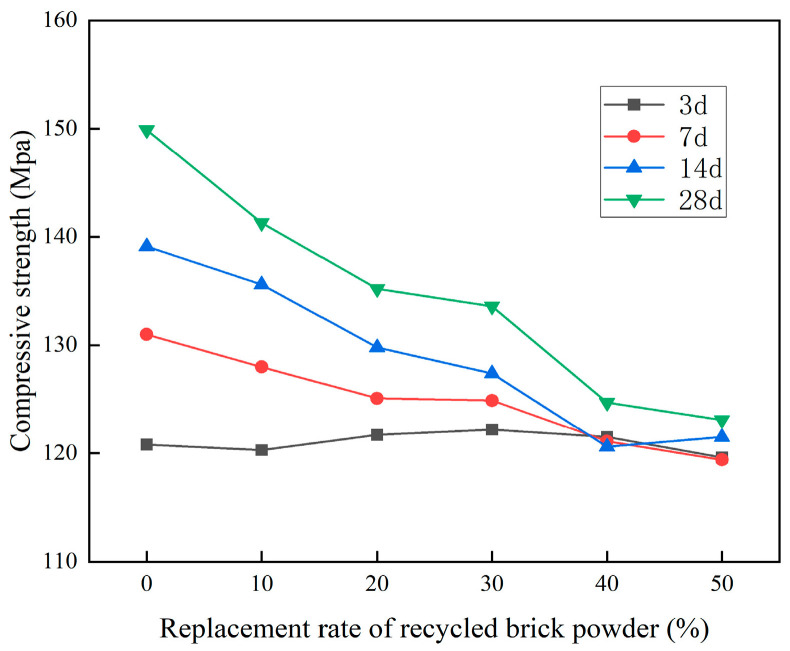
The compressive strength of recycled micropowder UHPC under different replacement rates.

**Figure 9 materials-16-01570-f009:**
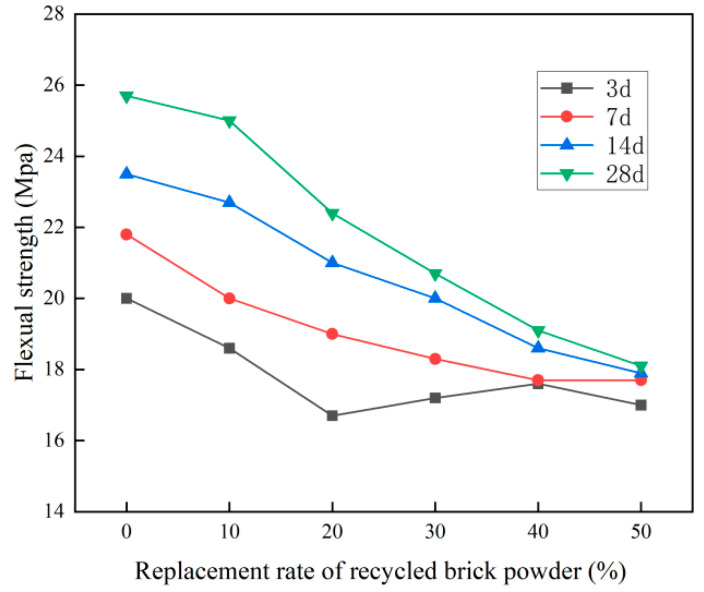
The flexural strength of the recycled micropowder UHPC with different substitution rates.

**Figure 10 materials-16-01570-f010:**
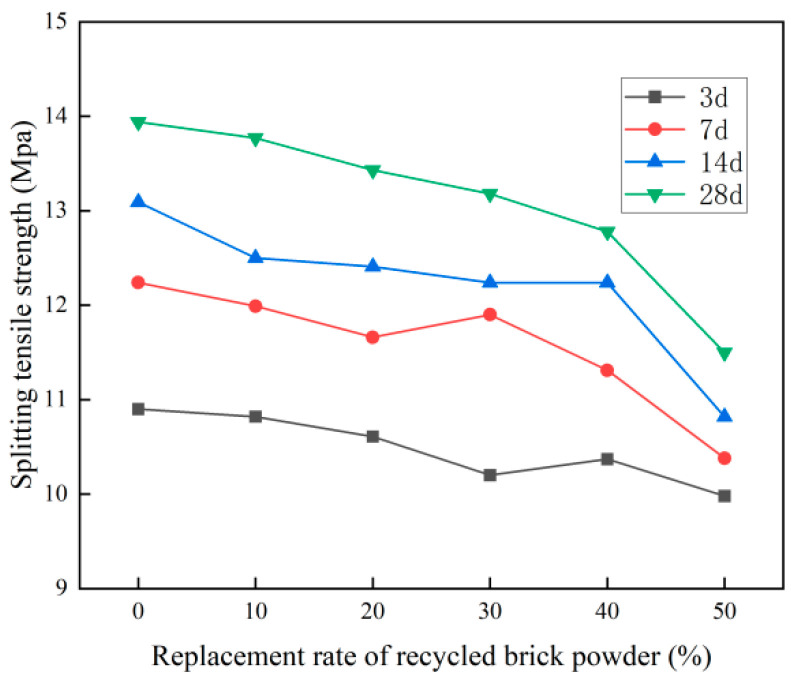
The splitting tensile strength of the recycled micropowder UHPC under different replacement rates.

**Figure 11 materials-16-01570-f011:**
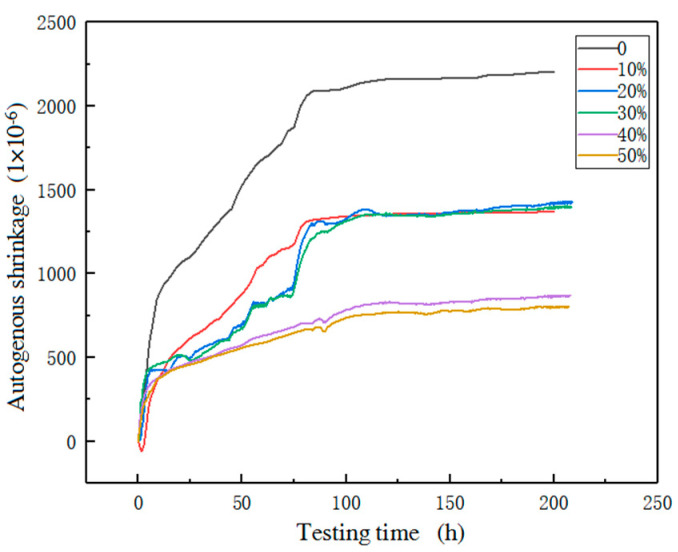
The early autogenous shrinkage curve of recycled brick micropowder UHPC.

**Figure 12 materials-16-01570-f012:**
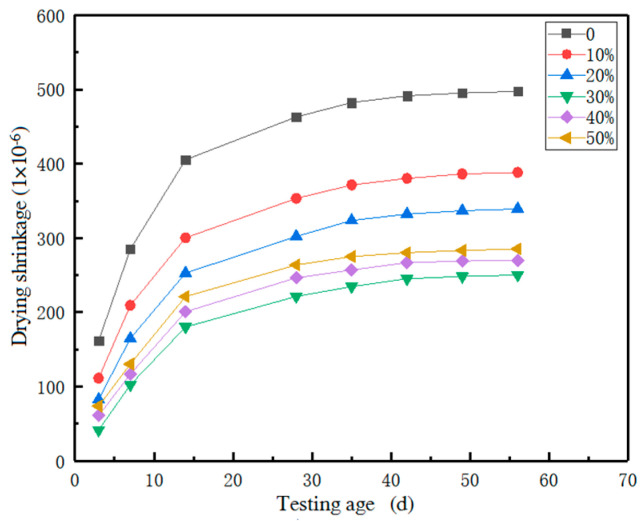
The drying shrinkage curve of the recycled brick powder UHPC.

**Figure 13 materials-16-01570-f013:**
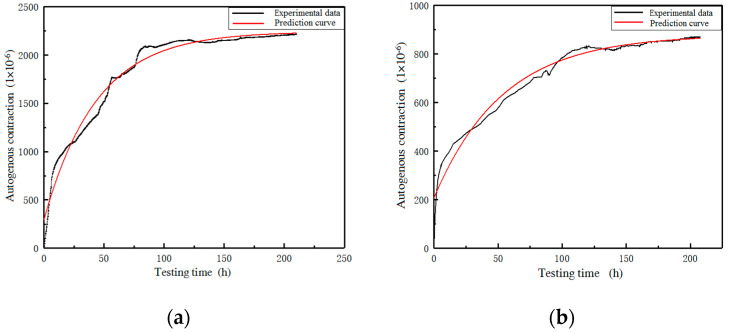
A comparison of the autogenous shrinkage test curve and prediction curve. (**a**) Baseline group; (**b**) 40% substitution rate group.

**Figure 14 materials-16-01570-f014:**
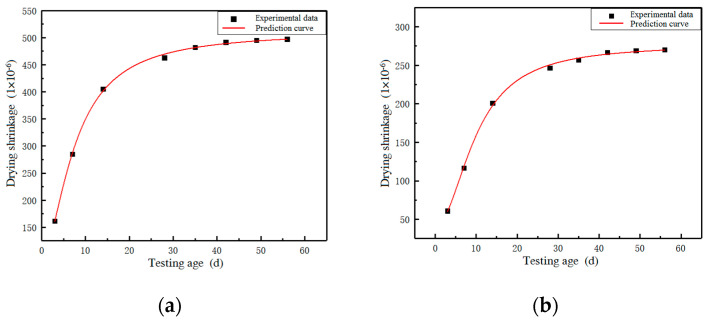
A comparison between the drying shrinkage test results and prediction curve. (**a**) Baseline group; (**b**) 40% substitution rate group.

**Table 1 materials-16-01570-t001:** The main technical indexes of cement.

Specific Surface Area/m^2^·kg^−1^	Stability	SO_3_/%	Cl^−^/%	MgO/%	Loss on Ignition/%	Setting Time/min		28 d Strength/MPa	
						Initial Set	Final Set	Flexural Strength	Compressive Strength
386	Qualified	2.38	0.046	3.93	3.06	150	200	8.9	63.4

**Table 2 materials-16-01570-t002:** The main material property indicators and components of recycled brick powder.

Bulk Density/kg·m^−3^	Specific Surface Area/m^2^·kg^−1^	Major Contents of Components/%					
		SiO_2_	Al_2_O_3_	Fe_2_O_3_	CaO	MgO	Na_2_O
993.8	600	68.15	16.51	7.20	1.80	0.94	0.65

**Table 3 materials-16-01570-t003:** Mix proportion of recycled brick powder UHPC (kg/m^3^).

Group	Replacement Rate(%)	Density/kg·m^−3^	Composition of Mixture/kg·m^−3^							
			Cement	Brick Powder	Fly Ash	Silica Fume	River Sand	Steel Fiber	Water Reducer	Water
R0	0	2356	700	0	100	200	1000	156	30	170
R1	10	2356	630	70	100	200	1000	156	30	170
R2	20	2356	560	140	100	200	1000	156	30	170
R3	30	2356	490	210	100	200	1000	156	30	170
R4	40	2356	420	280	100	200	1000	156	30	170
R5	50	2356	350	350	100	200	1000	156	30	170

Note: the volume content of steel fiber is 2%, which is converted to the mass of 156 kg/m^3^.

**Table 4 materials-16-01570-t004:** The effects of the replacement ratio of regenerated micropowder on compressive strength.

Replacement Rate (%)	Average Compressive Strength					
	3 d (MPa)	Standard Deviation	7 d (MPa)	14 d (MPa)	28 d (MPa)	Standard Deviation
0	120.8	3.78	131.0	139.1	149.9	6.77
10	120.3	3.56	128.0	135.6	141.3	6.12
20	121.7	3.12	125.1	129.8	135.2	5.77
30	122.2	3.87	124.9	127.4	133.6	5.54
40	121.5	3.67	121.1	120.6	124.7	4.34
50	119.6	3.22	119.4	121.5	123.1	4.29

**Table 5 materials-16-01570-t005:** The effects of the replacement ratio of regenerated micropowder on flexural strength.

Replacement Rate (%)	Average Flexural Strength					
	3 d (MPa)	Standard Deviation	7 d (MPa)	14 d (MPa)	28 d (MPa)	Standard Deviation
0	20.0	2.14	21.8	23.5	23.5	2.68
10	18.6	1.96	20.0	22.7	22.7	2.45
20	16.7	1.57	19.0	21.0	21.0	2.33
30	17.2	1.75	18.3	20.0	20.0	2.24
40	17.6	1.64	17.7	18.6	18.6	2.02
50	17.0	1.54	17.7	17.9	17.9	1.89

**Table 6 materials-16-01570-t006:** The effects of the replacement ratio of regenerated micropowder on splitting tensile strength.

Replacement Rate (%)	Average Splitting Tensile Strength					
	3 d (MPa)	Standard Deviation	7 d (MPa)	14 d (MPa)	28 d (MPa)	Standard Deviation
0	10.90	0.71	12.24	13.09	13.94	1.03
10	10.82	0.73	11.99	12.50	13.77	1.11
20	10.61	0.69	11.66	12.41	13.43	0.98
30	10.20	0.68	11.90	12.24	13.18	0.93
40	10.37	0.70	11.31	12.24	12.78	0.87
50	9.98	0.69	10.38	10.82	11.50	0.78

## Data Availability

The general data are included in the article. Additional data are available on request.

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
