# Peer review of "The Basic Mechanical Properties and Shrinkage Properties of Recycled Micropowder UHPC"

_materials, 2023, doi:10.3390/ma16041570_

Round 1

Reviewer 1 Report

Dear Authority,

The manuscript entitled ‘The Basic Mechanical Properties and Shrinkage Properties of  Recycled Micropowder UHPC’ suggest  the structural improvement such as mechanical and shrinkage properties in cement based materials by substituting cement with  clay brick powder

I think, the paper submits valuable information about variation in both mechanical and shrinkage properties Ultra-High-Performance Concrete (UHPC) by introducing recycled clay brick powder in different ratios. Overall, the data in manuscript is well represented and gives important information on Ultra-High-Performance Concrete (UHPC) explaining compressive, flexural, splitting tensile properties, autogenous shrinkage, and drying shrinkage properties of recycled brick powder.   However, there are few points requiring correction in   missing abstract and manuscript body. The manuscript needs to be revisited by considering following comments;

1- The abbreviation UHPC in abstract has to be state in long version as Ultra-High-Performance Concrete (UHPC) for easy of readership.

2- In experimental section, please state vacuum drying condition in terms of temperature. Is it room temperature?

3- Regarding xrd data analysis, after ball milling process, decrement in xrd peak reflection intensities is observed, please give more information with respect to changes in crystallite size or degree of crystallinity for recycled powder. For your convenience, you can use following manuscript;

a) https://doi.org/10.1016/j.mtcomm.2021.102202

b) https://doi.org/10.1016/j.mtcomm.2021.102637

4- The graphs used for compressive, flexural, and splitting tensile strength needs to modify because the y axis for all graphs are so wide. Make them narrower. For instance, for compressive strength illustration, set the y axis between 100 to 150 MPa instead of 0 -150 MPa.   

After minor modification, the paper could be considered for publication in Materials.

Best wishes,

Reviewer 2 Report

I thank the authors for the good job they have done with the evaluation of experiments and writing the paper.

Following suggestions and recommendations could be considered in the revised version of the paper.

1) I miss information on measuring the uncertainty of all presented results.  It must be completed.

2) Fig. 2 is useless.

3) Fig. 4 a) is misleading, it could be omitted and labeled as XRD analysis of activated recycled brick powder.

4) Fig. 5 presents well-known normalized tests; they could be omitted.

5) No comparison of obtained results with other previously published papers is given. It must be completed as it is significant information for verification and validation of presented data.

6) I miss information on basic material parameters, such as bulk density, open porosity, pore size, and distribution - the presented properties are very strongly dependent on the bulk density of prepared UHPC.

Round 2

Reviewer 2 Report

Based on the high quality of the paper I recommend its publication. I thank authors for really good job they have done within the evaluation of experiments and writing the paper.

Author Response

Thank the reviewers for their valuable comments, and thank you very much for your approval of the paper.